# Are Convex Optimization Curves Convex?

**Guy Barzilai**                                                    *guybarzilai1@mail.tau.ac.il*
*Tel-Aviv University*

**Ohad Shamir**                                                    *ohad.shamir@weizmann.ac.il*
*Weizmann Institute of Science*

**Moslem Zamani**                                                  *moslem.zamani@uclouvain.be*
*UCLouvain*

**Reviewed on OpenReview:** *https://openreview.net/forum?id=TZtpxselK2*

## Abstract

In this paper, we study when we might expect the optimization curve induced by gradient descent to be *convex* – precluding, for example, an initial plateau followed by a sharp decrease, making it difficult to decide when optimization should stop. Although such undesirable behavior can certainly occur when optimizing general functions, might it also occur in the benign and well-studied case of smooth convex functions? As far as we know, this question has not been tackled in previous work. We show, perhaps surprisingly, that the answer crucially depends on the choice of the step size. In particular, for the range of step sizes which are known to result in monotonic convergence to an optimal value, we characterize a regime where the optimization curve will be provably convex, and a regime where the curve can be non-convex. We also extend our results to gradient flow, and to the closely-related but different question of whether the gradient norm decreases monotonically.

## 1 Introduction

We consider the well-known gradient descent algorithm with constant step-size, which given a function $f : \mathbb{R}^n \to \mathbb{R}$, an initial point $x_0 \in \mathbb{R}^n$ and step-size parameter $\eta > 0$, generates a sequence of points $\{x_n\}_{n=0}^{\infty}$ following the recurrence relation

$$\forall n \in \mathbb{N} : x_n = x_{n-1} - \eta \nabla f(x_{n-1}) \ .$$

This sequence of points induces an *optimization curve*, which is the linear interpolation of $\{(n, f(x_n))\}_{n=0}^{\infty}$ in $\mathbb{R}^2$ (see Figure 1 for an example).

Our motivating question is to understand the properties of this optimization curve. For example, if the function $f$ is convex and smooth, then choosing the step size $\eta$ appropriately, it is well-known that the curve decreases monotonically to the minimal value of $f$ (Nesterov, 2018). However, other properties might also be of interest. For example, an undesirable phenomenon concerning the optimization curve is when it plateaus for a while, and after that substantially decreases. This phenomenon is undesirable, since it can cause the user to terminate optimization prematurely, thinking that a minimum has been reached, where in fact continuing the optimization would have resulted in a significant improvement. An important property that disallows this undesirable phenomenon is *convexity* of the optimization curve: Namely, that the slope of the curve (or equivalently, $\{f(x_n) - f(x_{n+1})\}_{n=0}^{\infty}$) decreases monotonically. Moreover, convexity of the optimization curve occurs very often in practice, in a wide variety of problems. However, to the best of our knowledge, no previous work attempted to rigorously study when we might expect the optimization curve to be convex.

To begin this investigation, we first note that for general non-convex functions, we have no hope of guaranteeing a convex optimization curve. For instance, if we initialize gradient descent close to a maximum of a

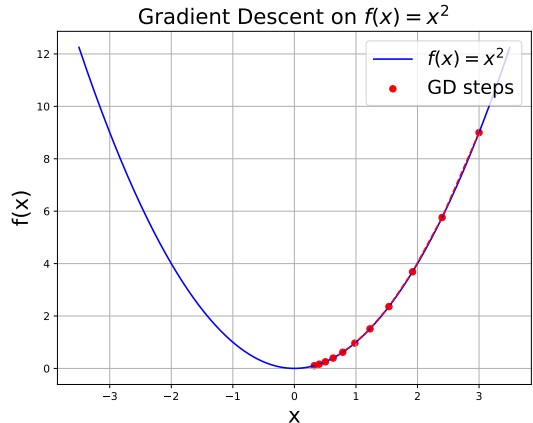 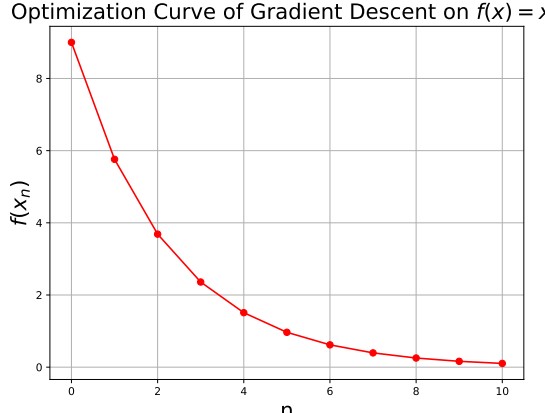

Figure 1: Gradient descent on $f(x) = x^2$ with step-size $\eta = 0.1$ and initial point $x_0 = 3$.

smooth locally concave function, it is easily seen that the optimization rate will increase over time, leading to a concave optimization curve. Moreover, even if we consider the more benign case of convex functions, it is well-known that the optimization curve of gradient descent may not monotonically decrease in general (and certainly not be convex), as shown in the following simple example:

**Example 1.** *Consider the convex 2-Lipschitz function*

$$f(x) = |x| + \max\{0, x\}$$

*on $\mathbb{R}$, which is minimized at $0$. For any step size $\eta > 0$, suppose we initialize gradient descent at $x_0 = -\frac{\eta}{4}$. By definition of gradient descent, we have*

$$x_1 = x_0 - \eta f'(x_0) = -\frac{\eta}{4} + \eta = \frac{3}{4}\eta \quad and \quad x_2 = x_1 - \eta f'(x_1) = \frac{3}{4}\eta - 2\eta = -\frac{5}{4}\eta \;,$$

*hence $f(x_0) = \frac{\eta}{4}$, $f(x_1) = \frac{3}{2}\eta$, $f(x_2) = \frac{5}{4}\eta$. Therefore, the optimization curve first increases and then decreases, and thus is neither convex nor monotonically decreasing.*

This observation motivates us in focusing on the class of $L$-smooth convex functions, where the gradient is $L$-Lipschitz. This is an extremely well-studied class for gradient descent procedures, and in particular, it is well-known that for any $\eta \in (0, \frac{2}{L})$, the optimization curve of gradient descent will monotonically converge to the minimal value (whereas for larger value of $\eta$, gradient descent may diverge; see Nesterov (2018, Theorem 2.1.14)). Thus, we will be interested in whether the optimization curve of gradient descent is convex, when optimizing such functions and in this step size regime. As far as we know, this question has not been addressed in previous literature.

In this paper, we provide an answer to this question, which is (perhaps unexpectedly) somewhat subtle: On the one hand, for any $\eta \in (0, \frac{1.75}{L}]$ (which includes the worst-case optimal step size $1/L$), we prove that the optimization curve is indeed convex. On the other hand, we prove that for $\eta \in (\frac{1.75}{L}, \frac{2}{L})$ the optimization curve may not be convex – even though the curve is still monotonically decreasing and gradient descent is guaranteed to converge to a minimal value. In other words, there exist step sizes that ensure monotonic convergence of gradient descent, but do not ensure convexity of the induced optimization curve.

As another contribution, we consider a closely-related but different question: Namely, whether the norm of the gradients $\{\|\nabla f(x_n)\|\}_{n=0}^{\infty}$ monotonically decrease, for $L$-smooth convex functions. This question is related, as the gradient captures the local slope of the function, which correlates with the magnitude of the objective function decrease after a single gradient step, assuming smoothness and that the step is small enough (in fact, this intuition is used more formally in our proofs). Although one might suspect that the behavior of gradient norm decay will be similar to the question of optimization curve convexity, we prove

that the answer here is different and more positive: The gradient norm is decreasing, for *any* $\eta \in (0, \frac{2}{L}]$. Finally, we extend our analysis to gradient flow (the continuous-time analogue of gradient descent), and prove that for smooth convex functions, the gradient flow optimization curve is always convex, and that the gradient norm is monotonically decreasing.

## 2 Preliminaries

We will need the following simple lemma, which essentially states that the integral of a non-decreasing function is convex:

**Lemma 1.** *Let $g : \mathbb{R} \to \mathbb{R}$ be a Riemann integrable function on every closed sub-interval of $\mathbb{R}$. Define $f : \mathbb{R} \to \mathbb{R}$ such that for every $x \in \mathbb{R}$:*

$$f(x) = \int_0^x g(t)dt.$$

*If $g$ is non-decreasing, then $f$ is convex.*

*Proof.* Let $s < u < t$ be real numbers. By definition of convexity, it is enough to show that

$$\frac{f(u) - f(s)}{u - s} \leq \frac{f(t) - f(u)}{t - u} \tag{1}$$

We note that by definition of $f$, it holds that $f(u) - f(s) = \int_s^u g(x)dx$, and by the monotonicity of $g$, it therefore holds that $(u - s) \cdot g(s) \leq f(u) - f(s) \leq (u - s) \cdot g(u)$. Thus,

$$g(s) \leq \frac{f(u) - f(s)}{u - s} \leq g(u) \quad \text{and similarly} \quad g(u) \leq \frac{f(t) - f(u)}{u - t} \leq g(t) . \tag{2}$$

Obviously, (2) implies (1), completing the proof. □

As discussed in the introduction, we consider an optimization curve induced $\{f(x_n)\}_{n=0}^{\infty}$ to be convex, if the curve formed from the linear interpolation of the discrete points $\{(n, f(x_n))\}_{n=0}^{\infty}$ is convex (in the standard sense, e.g. its epigraph is a convex set). This is formalized in the following definition:

**Definition 1.** *Let $\{a_n\}_{n=0}^{\infty}$ be a sequence of real numbers. Then we say that the mapping $n \mapsto a_n$ is convex over $\mathbb{Z}_{\geq 0}$, if the function $g : [0, \infty) \to \mathbb{R}$ defined as*

$$g(t) = a_{\lfloor t \rfloor} + (t - \lfloor t \rfloor)(a_{\lfloor t \rfloor + 1} - a_{\lfloor t \rfloor})$$

*is convex.*

By Lemma 1 and the Newton-Leibniz theorem (as appears, for instance, in Bartle & Sherbert (2011, Theorem 7.3.1)), we see that a real valued sequence $\{a_n\}_{n=0}^{\infty}$ is convex over $\mathbb{Z}_{\geq 0}$ if and only if the sequence $\{a_n - a_{n+1}\}_{n=0}^{\infty}$ is non-increasing (actually, the "only if" part follows from elementary properties of convex functions). It shall often be convenient to use the latter characterization rather than the former.

## 3 Gradient Descent

In this section, we shall investigate the optimization curves induced by gradient descent on convex $L$-smooth functions. For such functions, it is well-known that for any $\eta \in (0, \frac{2}{L})$, the gradient descent optimization curve monotonically decreases to the minimal function value (see Nesterov (2018, Theorem 2.1.14)). In the following theorem, we prove that in the more restricted regime $\eta \in (0, \frac{1.75}{L}]$, the optimization curve is also convex:

**Theorem 1.** *Let $f : \mathbb{R}^n \to \mathbb{R}$ be a convex $L$-smooth function, and let $\{x_n\}_{n=0}^{\infty}$ be the sequence of iterates produced by gradient descent, starting from some $x_0 \in \mathbb{R}^n$, using a step size $\eta \in (0, \frac{1.75}{L}]$. Then the mapping $n \mapsto f(x_n)$ is convex (equivalently, the sequence $\{f(x_n) - f(x_{n+1})\}_{n=0}^{\infty}$ is non-increasing).*

*Proof.* Arguing by induction and using the fact that $x_0$ is arbitrary, it suffices to prove that

$$f(x_2) - f(x_1) \geq f(x_1) - f(x_0) \,. \tag{3}$$

By convexity and by $L$-smoothness of $f$, we have (see Nesterov (2018, Theorem 2.1.5)):

$$f(x_0) - f(x_1) \geq \langle \nabla f(x_1), x_0 - x_1 \rangle + \frac{1}{2L} \|\nabla f(x_1) - \nabla f(x_0)\|^2 \,,$$

$$f(x_2) - f(x_1) \geq \langle \nabla f(x_1), x_2 - x_1 \rangle + \frac{1}{2L} \|\nabla f(x_2) - \nabla f(x_1)\|^2 \,,$$

$$f(x_2) - f(x_0) \geq \langle \nabla f(x_0), x_2 - x_0 \rangle + \frac{1}{2L} \|\nabla f(x_2) - \nabla f(x_0)\|^2 \,.$$

Due to gradient descent update rule, we have $x_1 = x_0 - \eta \nabla f(x_0)$ and $x_2 = x_0 - \eta \nabla f(x_0) - \eta \nabla f(x_1)$ . Substituting these values into the inequalities, we obtain:

$$f(x_0) - f(x_1) \geq \eta \langle \nabla f(x_1), \nabla f(x_0) \rangle + \frac{1}{2L} \|\nabla f(x_1) - \nabla f(x_0)\|^2 \,,$$

$$f(x_2) - f(x_1) \geq -\eta \|\nabla f(x_1)\|^2 + \frac{1}{2L} \|\nabla f(x_2) - \nabla f(x_1)\|^2 \,,$$

$$f(x_2) - f(x_0) \geq -\eta \|\nabla f(x_0)\|^2 - \eta \langle \nabla f(x_0), \nabla f(x_1) \rangle + \frac{1}{2L} \|\nabla f(x_2) - \nabla f(x_0)\|^2 \,.$$

Multiplying the above inequalities by $\frac{3}{2}$, $\frac{1}{2}$ and $\frac{1}{2}$ respectively and summing them up, we obtain:

$$\begin{aligned} f(x_2) - f(x_1) - (f(x_1) - f(x_0)) \geq & \frac{3\eta}{2} \langle \nabla f(x_1), \nabla f(x_0) \rangle + \frac{3}{4L} \|\nabla f(x_1) - \nabla f(x_0)\|^2 \\ & - \frac{\eta}{2} \|\nabla f(x_1)\|^2 + \frac{1}{4L} \|\nabla f(x_2) - \nabla f(x_1)\|^2 \\ & - \frac{\eta}{2} \|\nabla f(x_0)\|^2 - \frac{\eta}{2} \langle \nabla f(x_0), \nabla f(x_1) \rangle \\ & + \frac{1}{4L} \|\nabla f(x_2) - \nabla f(x_0)\|^2 \,. \end{aligned} \tag{4}$$

Rearranging terms, the right-hand side equals

$$\left( \frac{3}{4L} - \frac{\eta}{2} \right) \|\nabla f(x_1) - \nabla f(x_0)\|^2 + \frac{1}{4L} \|\nabla f(x_2) - \nabla f(x_1)\|^2 + \frac{1}{4L} \|\nabla f(x_2) - \nabla f(x_0)\|^2$$

$$= \left( \frac{7}{8L} - \frac{\eta}{2} \right) \|\nabla f(x_1) - \nabla f(x_0)\|^2$$

$$+ \frac{1}{4L} \left( -\frac{1}{2} \|\nabla f(x_1) - \nabla f(x_0)\|^2 + \|\nabla f(x_2) - \nabla f(x_1)\|^2 + \|\nabla f(x_2) - \nabla f(x_0)\|^2 \right)$$

$$= \left( \frac{7}{8L} - \frac{\eta}{2} \right) \|\nabla f(x_1) - \nabla f(x_0)\|^2 + \frac{1}{2L} \left\| \nabla f(x_2) - \frac{1}{2} \nabla f(x_1) - \frac{1}{2} \nabla f(x_0) \right\|^2 \,,$$

where the last transition is by the elementary equality $-\frac{1}{2} \|b - a\|^2 + \|c - b\|^2 + \|c - a\|^2 = 2 \left\| c - \frac{1}{2}b - \frac{1}{2}a \right\|^2$, which can be easily verified via expansion. Overall, we have shown that

$$\begin{aligned} f(x_2) - f(x_1) - (f(x_1) - f(x_0)) \geq & \left( \frac{7}{8L} - \frac{\eta}{2} \right) \|\nabla f(x_1) - \nabla f(x_0)\|^2 \\ & + \frac{1}{2L} \left\| \nabla f(x_2) - \frac{1}{2} \nabla f(x_1) - \frac{1}{2} \nabla f(x_0) \right\|^2 \,. \end{aligned}$$

The first term is non-negative (since we assume $\eta \leq \frac{7}{4L} = \frac{1.75}{L}$), and so is the second term. This implies (3) and concludes the proof. $\qquad \square$

We note that the theorem applies for step sizes in the range $(0, \frac{1.75}{L}]$, which is more restrictive than the regime $(0, \frac{2}{L})$ where gradient descent is known to monotonically decrease to an optimal value. Thus, one may wonder whether this restriction is necessary, or perhaps Theorem 1 can be extended for a broader range of step-sizes. In the following theorem, we prove that precisely this restriction is indeed necessary – namely, for any step size $\eta \in (\frac{1.75}{L}, \frac{2}{L})$, gradient descent will converge and the optimization curve will monotonically decrease, without necessarily being convex:

**Theorem 2.** *For every $L > 0$ there exists a convex $L$-smooth function $f : \mathbb{R} \to \mathbb{R}$ and $x_0 \in \mathbb{R}$ such that for every step-size $\eta \in (\frac{1.75}{L}, \frac{2}{L})$, the mapping $n \mapsto f(x_n)$ is* not *convex.*

*Proof.* We note that without loss of generality, it suffices to prove the theorem for $L = 1$. To see this, suppose our theorem holds for $L = 1$. This means that there exists a convex 1-smooth function $f : \mathbb{R} \to \mathbb{R}$ and $x_0 \in \mathbb{R}$, such that for every step-size $\eta \in (1.75, 2)$, the optimization curve of gradient descent on $f$ with initial point $x_0$ and a constant step-size $\eta$ isn't convex. Now, define $g : \mathbb{R} \to \mathbb{R}$ such that for every $t \in \mathbb{R}$: $g(t) = f(\sqrt{L}t)$, define $\tilde{x}_0 = \frac{x_0}{\sqrt{L}}$ and define $\tilde{\eta} = \frac{\eta}{L}$. The reader may verify that $g$ is convex and $L$-smooth. Now, for $n \in \mathbb{N}$, define

$$x_n = x_{n-1} - \eta f'(x_{n-1}), \;\; \tilde{x}_n = \tilde{x}_{n-1} - \tilde{\eta} g'(\tilde{x}_{n-1}) \; .$$

Arguing by induction, we claim that for $n \in \mathbb{Z}_{\geq 0}$ it holds that $\tilde{x}_n = \frac{x_n}{\sqrt{L}}$. Indeed, the base case $n = 0$ holds by definition, and if the claim holds for $n \in \mathbb{Z}_{\geq 0}$, then

$$\tilde{x}_{n+1} = \tilde{x}_n - \tilde{\eta} g'(\tilde{x}_n) = \frac{x_n}{\sqrt{L}} - \frac{\eta}{L} \sqrt{L} f'(\sqrt{L} \frac{x_n}{\sqrt{L}})$$

$$= \frac{1}{\sqrt{L}}(x_n - \eta f'(x_n)) = \frac{x_{n+1}}{\sqrt{L}} \; .$$

It obviously follows by the definition of $g$ that $g(\tilde{x}_n) = f(x_n)$ for all $n \in \mathbb{Z}_{\geq 0}$. Thus, the optimization curve corresponding to $g, \tilde{x}_0$ and $\tilde{\eta}$ is the same as the optimization curve corresponding to $f, x_0$ and $\eta$ - which shows why it suffices to prove the theorem for $L = 1$.

We now turn to prove the theorem for $L = 1$. Specifically, consider the function $f : \mathbb{R} \to \mathbb{R}$ defined as

$$f(t) = \begin{cases} \frac{1}{2}t^2, & t \leq 1 \\ t - \frac{1}{2}, & t > 1 \; . \end{cases}$$

It is fairly obvious and easy to check that $f$ is indeed a convex 1-smooth function.

Now, set $x_0 = -1.8$ and let there be $\eta \in (1.75, 2)$. We have

$$x_1 = x_0 - \eta f'(x_0) = (\eta - 1)(-x_0) = 1.8(\eta - 1) \; ,$$

and due to the fact that $\eta > 1.75$, we have $x_1 > 0.75 \cdot 1.8 = 1.35$. This implies that

$$x_2 = x_1 - \eta f'(x_1) = x_1 - \eta = -1.8 + 0.8\eta < -1.8 + 0.8 \cdot 2 = -0.2 \; ,$$

where in the last equality we recalled that $\eta < 2$. Thus:

$$f(x_0) - f(x_1) < f(x_1) - f(x_2) \iff \frac{1}{2}x_0^2 - (x_1 - \frac{1}{2}) < (x_1 - \frac{1}{2}) - \frac{1}{2}x_2^2$$

$$\iff 0.5(-1.8)^2 - (1.8\eta - 1.8 - 0.5) < (1.8\eta - 1.8 - 0.5) - 0.5(-1.8 + 0.8\eta)^2 \; .$$

Rearranging the terms, we get the equivalent inequality $\eta^2 - 15.75\eta + 24.5 < 0$, which is easily verified to be equivalent to $1.75 < \eta < 14$. This holds due to the assumption that $\eta \in (1.75, 2)$. Thus, $f(x_0) - f(x_1) < f(x_1) - f(x_2)$ holds, which means that the relevant optimization curve is not convex. This concludes our proof. □

So far, we have considered the convexity of the optimization curve induced by gradient descent. A related property of interest is how the sequence of gradient norms $\{\|\nabla f(x_n)\|\}_{n=0}^{\infty}$ behave. It is well-known (see Nesterov (2012)) that for convex smooth functions, the *minimal* value after $N$ iterations, namely $\min_{n \leq N} \|\nabla f(x_n)\|$, is upper bounded by a function monotonically decreasing to 0 (as should be expected when the method converges to a global minimum). However, does the sequence $\{\|\nabla f(x_n)\|\}_{n=0}^{\infty}$ itself decay monotonically? The following theorem establishes that this is true, for any $\eta$ in the regime where gradient descent converges:

**Theorem 3.** *Let $f : \mathbb{R}^n \to \mathbb{R}$ be a convex $L$-smooth function, and let $\{x_n\}_{n=0}^{\infty}$ be the sequence of iterates produced by gradient descent, starting from some $x_0 \in \mathbb{R}^n$, using a step size $\eta \in (0, \frac{2}{L}]$. Then the sequence $\{\|\nabla f(x_n)\|\}_{n=0}^{\infty}$ is non-increasing.*

*Proof.* Arguing by induction, and using the fact that $x_0$ is arbitrary, it suffices to show that $\|\nabla f(x_0)\| \geq \|\nabla f(x_1)\|$.

Using a standard inequality for convex $L$-smooth functions (see Nesterov (2018, Theorem 2.1.5)), we have that for all $x, y \in \mathbb{R}^n$:

$$\langle \nabla f(x) - \nabla f(y), x - y \rangle \geq \frac{1}{L} \|\nabla f(x) - \nabla f(y)\|^2 \ . \tag{5}$$

Plugging $x_0, x_1$ into (5), we have

$$\langle \nabla f(x_1) - \nabla f(x_0), x_1 - x_0 \rangle \geq \frac{1}{L} \|\nabla f(x_1) - \nabla f(x_0)\|^2$$

$$\iff \langle \nabla f(x_1) - \nabla f(x_0), -\eta \nabla f(x_0) \rangle \geq \frac{1}{L} \|\nabla f(x_1) - \nabla f(x_0)\|^2$$

$$\iff -\eta \left( \langle \nabla f(x_1), \nabla f(x_0) \rangle - \|\nabla f(x_0)\|^2 \right) \geq \frac{1}{L} \left( \|\nabla f(x_1)\|^2 - 2 \langle \nabla f(x_1), \nabla f(x_0) \rangle + \|\nabla f(x_0)\|^2 \right)$$

$$\iff (\frac{2}{L} - \eta) \langle \nabla f(x_1), \nabla f(x_0) \rangle \geq (\frac{1}{L} - \eta) \|\nabla f(x_0)\|^2 + \frac{1}{L} \|\nabla f(x_1)\|^2 \ . \tag{6}$$

Now, if $\eta = \frac{2}{L}$, then by substituting for $\eta$ and rearranging the terms, we get from (6) that

$$\frac{1}{L} \|\nabla f(x_0)\|^2 \geq \frac{1}{L} \|\nabla f(x_1)\|^2 \ ,$$

which obviously implies the desired result. Otherwise, we have $\frac{2}{L} - \eta > 0$ by assumption. Using Cauchy-Schwarz, (6) implies

$$(\frac{2}{L} - \eta) \|\nabla f(x_1)\| \cdot \|\nabla f(x_0)\| \geq (\frac{1}{L} - \eta) \|\nabla f(x_0)\|^2 + \frac{1}{L} \|\nabla f(x_1)\|^2 \ .$$

Rearranging the terms, we get the equivalent inequality

$$(\|\nabla f(x_0)\| - \|\nabla f(x_1)\|) \cdot \left( (\eta - \frac{1}{L}) \|\nabla f(x_0)\| + \frac{1}{L} \|\nabla f(x_1)\| \right) \geq 0 \ . \tag{7}$$

Now, if $\|\nabla f(x_1)\| = 0$ then we are obviously done. Otherwise, assume for the sake of contradiction that $\|\nabla f(x_1)\| > \|\nabla f(x_0)\|$. Then, splitting to the cases $\eta \in \left(0, \frac{1}{L}\right)$ and $\eta \in \left[\frac{1}{L}, \frac{2}{L}\right)$, it is easy to verify that

$$\left( (\eta - \frac{1}{L}) \|\nabla f(x_0)\| + \frac{1}{L} \|\nabla f(x_1)\| \right) > 0 \ .$$

Together with $\|\nabla f(x_0)\| - \|\nabla f(x_1)\| < 0$ this results in

$$(\|\nabla f(x_0)\| - \|\nabla f(x_1)\|) \cdot \left( (\eta - \frac{1}{L}) \|\nabla f(x_0)\| + \frac{1}{L} \|\nabla f(x_1)\| \right) < 0 \ ,$$

which contradicts (7).

Therefore, in any case, we have $\|\nabla f(x_0)\| \geq \|\nabla f(x_1)\|$ as required. $\qquad \square$

## 4 Gradient Flow

In this section we extend our results to the optimization curve induced by gradient flow, and prove that it is always convex (when optimizing smooth convex functions).

First, we recall that given a function $f \in C^1(\mathbb{R}^n, \mathbb{R})$ (namely, differentiable with a continuous gradient), and some $x_0 \in \mathbb{R}^n$, the gradient flow $(x(t))_{t \geq 0}$ is defined by the differential equation

$$\begin{cases} x'(t) = -\nabla f(x(t)) \\ x(0) = x_0 \ . \end{cases}$$

Assuming $L$-smoothness of $f$ for some $L < \infty$ is a sufficient condition for this differential equation to have a unique, well-defined solution globally on all of $t \geq 0$ (this follows from the Picard-Lindelöf theorem). We can then define the optimization curve induced by gradient flow as follows:

**Definition 2.** *Given a function $f \in C^1(\mathbb{R}^n, \mathbb{R})$ for which gradient flow (starting from some $x_0$) is uniquely defined, the associated optimization curve $g : [0, \infty) \to \mathbb{R}$ is defined as $g(t) = f(x(t))$ for all $t \in [0, \infty)$.*

Before discussing the case of smooth convex functions, let us first consider the simpler case of convex $C^2$ functions. In that case, one may verify that there exists a unique solution defined for $t \geq 0$ for the gradient flow differential equation (again, this can be shown to follow from the Picard-Lindelöf theorem and the fact that the gradient norm is decreasing, due to the function's smoothness and convexity). Moreover, in that case convexity of the optimization curve is easy to establish:

**Theorem 4.** *Let $f : \mathbb{R}^n \to \mathbb{R}$ be a convex function such that $f \in C^2(\mathbb{R}^n, \mathbb{R})$ and $x_0 \in \mathbb{R}^n$. Then the resulting gradient flow optimization curve is convex.*

*Proof.* Recall that the optimization curve is defined as $g(t) = f(x(t))$ for all $t \geq 0$. By elementary results, a sufficient condition for convexity of $g$ is that it is twice differentiable, with a non-negative second derivative.

By the definition of the gradient flow differential equation, we have for all $t \geq 0$:

$$x'(t) = -\nabla f(x(t)) \ . \tag{8}$$

Therefore, $x(\cdot)$ is differentiable for all $t \geq 0$. Now, using the chain rule we have:

$$g'(t) = \langle \nabla f(x(t)), x'(t) \rangle \overset{(8)}{=} -\|\nabla f(x(t))\|^2 \ . \tag{9}$$

Owing to the fact that $f \in C^2(\mathbb{R}^n, \mathbb{R})$, we can differentiate (9) again using the chain rule to get:

$$g''(t) = -2 \langle \nabla f(x(t)), H_f(x(t)) x'(t) \rangle \overset{(8)}{=} 2 \langle \nabla f(x(t)), H_f(x(t)) \nabla f(x(t)) \rangle \tag{10}$$

where $H_f$ is the hessian matrix of $f$. We note that $f$ is convex and $f \in C^2(\mathbb{R}^n, \mathbb{R})$. Therefore, for all $z \in \mathbb{R}^n$ it hold that $H_f(z)$ is positive semi-definite (see Nesterov (2018, Theorem 2.1.4)). Specifically, for all $t \geq 0$,

$$(\nabla f(x(t)))^T H_f(x(t)) \nabla f(x(t)) \geq 0$$

and therefore by (10), $g''(t) \geq 0$ for all $t \geq 0$. This implies that $g$ is twice differentiable on $[0, \infty)$, and that its second derivative is non-negative, which concludes the proof. $\square$

We now turn to discuss the case of convex and smooth functions (which are not necessarily twice-differentiable). The theorem below implies that the resulting optimization curve is always convex:

**Theorem 5.** *Let $f : \mathbb{R}^n \to \mathbb{R}$ be a convex $L$-smooth function for some $L \in (0, \infty)$, and $x_0 \in \mathbb{R}^n$. Then the resulting gradient flow optimization curve is convex.*

The proof of the theorem (which is a bit involved and appears in the appendix) builds on a variant of the proof of Theorem 1, and the fact that gradient descent can be seen as a discretization of gradient flow.

Furthermore, from Theorem 5, one concludes the gradient flow equivalent of Theorem 3.

**Theorem 6.** *Let $f : \mathbb{R}^n \to \mathbb{R}$ be a convex $L$-smooth function for some $L \in (0, \infty)$, and $x_0 \in \mathbb{R}^n$. Let $x(\cdot)$ be the unique solution for the corresponding gradient flow differential equation. Then the map $t \mapsto \|\nabla f(x(t))\|$ is non-increasing over $[0, \infty)$*

*Proof.* Recall that the optimization curve is defined as $g(t) = f(x(t))$ for all $t \geq 0$. From Theorem 5, $g$ is convex, and by the monotonicity of gradients of convex functions, it follows that $g'$ is non-decreasing over $[0, \infty)$. From the definition of the gradient flow differential equation, for $t \in [0, \infty)$ it holds that:

$$x'(t) = -\nabla f(x(t)) \ .$$

Therefore, by the chain rule, for $t \in [0, \infty)$:

$$g'(t) = -\|\nabla f(x(t))\|^2 \ .$$

Hence, it follows that $t \mapsto -\|\nabla f(x(t))\|^2$ is non-decreasing on $t \in [0, \infty)$, and the result follows. $\square$

## 5 Conclusion and Discussion

In this paper, we studied the convexity of optimization curves induced by gradient descent, focusing on convex $L$-smooth functions. For step sizes that assure monotonic convergence – merely $\eta \in (0, \frac{2}{L})$, we identified the regime that ensures convexity of the induced optimization curve, namely $\eta \in (0, \frac{1.75}{L}]$. Moreover, we showed that in the complementary regime of $(\frac{1.75}{L}, \frac{2}{L})$, the induced optimization curve may not be convex, even though the gradient descent still converges monotonically to a minimum. In addition, we used the close connection between gradient descent and gradient flow to show that the optimization curve induced by gradient flow is convex as well. Finally, we studied the closely-related question of whether the gradient norms decrease, and showed that here there are no separate regimes: For both gradient descent and gradient flow, the gradient norms decrease monotonically.

Our work leaves several open questions and directions for future research. For example, the proof of non-convexity in Theorem 2 relies on showing lack of convexity between two consecutive steps - i.e, we showed that $f(x_0) - f(x_1) < f(x_1) - f(x_2)$. One may ask whether there exist cases where more steps fail to be convex - i.e, $f(x_n) - f(x_{n+1}) < f(x_{n+1}) - f(x_{n+2})$ holds for multiple $n \in \mathbb{Z}_{\geq 0}$, or whether the optimization curve might be concave for several consecutive steps. Another obvious direction for research is to consider other optimization settings where the optimization curve is convex or non-convex, or consider this question for other optimization methods.

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

# A  Proof of Theorem 5

In order to prove theorem 5, we recall the well-known fact that gradient descent can be seen as a discrete approximation of gradient flow, namely the solution of its differential equation:

**Definition 3.** *Let $f \in C^1(\mathbb{R}^n, \mathbb{R})$ be a convex function, and $x_0 \in \mathbb{R}^n$ such that the gradient flow differential equation has a unique solution $x(\cdot)$ (as a function on $[0, \infty)$). Let $\eta > 0$ be fixed.*
*As in gradient descent, for $n \in \mathbb{N}$ define: $x_n = x_{n-1} - \eta \nabla f(x_{n-1})$.*
*Define Euler's approximation with step-size $\eta$ to be $x^{(\eta)} : [0, \infty) \to \mathbb{R}^n$ such that for all $t \in [0, \infty)$,*

$$x^{(\eta)}(t) = x_{\left\lfloor \frac{t}{\eta} \right\rfloor} - \left( t - \left\lfloor \frac{t}{\eta} \right\rfloor \eta \right) \nabla f \left( x_{\left\lfloor \frac{t}{\eta} \right\rfloor} \right)$$

The reader may have already observed that in fact, the points generated by gradient descent with step-size $\eta > 0$ are exactly the points $\left\{ x^{(\eta)}(n\eta) \right\}_{n=0}^{\infty}$ where $x^{(\eta)}(\cdot)$ is the corresponding Euler's approximation. In other words - gradient descent with step-size $\eta > 0$, is the Euler discretization (with step-size $\eta$) associated with the gradient flow differential equation.

Another important property we will need is that as $\eta$ converges to 0, the Euler approximation generally converges to the solution of the differential equation. Proofs of this fact may be found in most textbooks on differential equations. However, for the sake of completeness we shall present such a proof, adapted to our specific assumptions and needs, and similar to the one that appears in Iserles (1996, Theorem 1.1):

**Theorem 7.** *Let $L > 0$, $f : \mathbb{R}^n \to \mathbb{R}$ be a convex $L$-smooth function and $x_0 \in \mathbb{R}^n$. For every $\eta > 0$ let $x^{(\eta)}(\cdot)$ be the corresponding Euler's approximation, and let $x(\cdot)$ be the unique solution for the corresponding gradient flow differential equation. Then for every $R > 0$*

$$x^{(\eta)} \xrightarrow{\eta \to 0} x$$

*uniformly on $[0, R]$.*

*Proof.* Fix $R > 0$ and $0 < \eta < 1$. $f$ is $L$-smooth, and thus $x(\cdot)$ (as a solution to the corresponding gradient flow differential equation) is uniquely defined on $[0, \infty)$. Moreover, from the definition of gradient flow, for all $t \in [0, \infty)$ it holds that:

$$x'(t) = -\nabla f(x(t)) .$$

Therefore, $x(\cdot)$ is differentiable on $[0, \infty)$, and therefore continuous on $[0, \infty)$. Now, by the fact that $f$ is $L$-smooth (and particularly $\nabla f \in C(\mathbb{R}^n, \mathbb{R}^n)$), it holds that the map $t \mapsto -\nabla f(x(t))$ is continuous on $[0, \infty)$, meaning that $x \in C^1([0, \infty), \mathbb{R}^n)$ (and particularly $x \in C([0, \infty), \mathbb{R}^n)$). Therefore, $x(\cdot)$ is bounded on $[0, R+1]$ - there exists $M > 0$ such that $\|x(t)\| \le M$ for all $t \in [0, R+1]$.

Now, from the fact that $\nabla f \in C(\mathbb{R}^n, \mathbb{R}^n)$, it holds that $\nabla f$ is bounded on $\{w \in \mathbb{R}^n : \|w\| \le M\}$. It therefore holds that $x(\cdot)$ has a bounded derivative on $[0, R+1]$, and therefore is Lipschitz on $[0, R+1]$ - i.e there exists $K > 0$ such that for all $t_1, t_2 \in [0, R+1]$,

$$\|x(t_1) - x(t_2)\| \le K |t_1 - t_2| .$$

Now, for $n \in \mathbb{Z}_{\ge 0}$ denote $e_n = x^{(\eta)}(n\eta) - x(n\eta)$. By definition of the differential equation and of Euler's approximation, it holds that $e_0 = 0$. Denote $n^* = \left\lfloor \frac{R}{\eta} \right\rfloor + 1$. Because $0 < \eta < 1$, it holds that $\eta \cdot n^* \le R+1$. Fixing some $n \in \{0, \ldots, n^* - 1\}$, we have

$$e_{n+1} = x^{(\eta)}((n+1)\eta) - x((n+1)\eta) = \left( x^{(\eta)}(n\eta) - \eta \nabla f(x^{(\eta)}(n\eta)) \right) - \left( x(n\eta) - \int_{n\eta}^{(n+1)\eta} \nabla f(x(s)) ds \right)$$

$$= e_n + \int_{n\eta}^{(n+1)\eta} \nabla f(x(s)) ds - \eta \nabla f(x^{(\eta)}(n\eta)) = e_n + \int_{n\eta}^{(n+1)\eta} \left[ \nabla f(x(s)) - \nabla f(x^{(\eta)}(n\eta)) \right] ds ,$$

and so

$$\|e_{n+1}\| \leq \|e_n\| + \int_{n\eta}^{(n+1)\eta} \left\| \nabla f(x(s)) - \nabla f(x^{(\eta)}(n\eta)) \right\| ds \leq \|e_n\| + L \int_{n\eta}^{(n+1)\eta} \left\| x(s) - x^{(\eta)}(n\eta) \right\| ds$$

$$\leq \|e_n\| + L \int_{n\eta}^{(n+1)\eta} \left( \|x(s) - x(n\eta)\| + \left\| x(n\eta) - x^{(\eta)}(n\eta) \right\| \right) ds$$

$$= (1 + L\eta) \|e_n\| + L \int_{n\eta}^{(n+1)\eta} \|x(s) - x(n\eta)\| \, ds$$

$$\leq (1 + L\eta) \|e_n\| + LK \int_{n\eta}^{(n+1)\eta} |s - n\eta| ds$$

$$= (1 + L\eta) \|e_n\| + \frac{LK}{2}\eta^2 \ .$$

Thus, arguing by induction, we get that for $n \in \{0, \ldots, n^*\}$,

$$\|e_n\| \leq \frac{LK}{2}\eta^2 \sum_{j=0}^{n-1} (1 + L\eta)^j \ .$$

Furthermore, a similar calculation implies that in fact, for every $t \in [0, R]$,

$$\left\| x^{(\eta)}(t) - x(t) \right\| \leq (1 + L\eta) \left\| e_{\lfloor \frac{t}{\eta} \rfloor} \right\| + \frac{LK}{2}\eta^2 \ ,$$

and thus for every $t \in [0, R]$,

$$\left\| x^{(\eta)}(t) - x(t) \right\| \leq \frac{LK}{2}\eta^2 \sum_{j=0}^{\lfloor \frac{t}{\eta} \rfloor} (1 + L\eta)^j \leq \frac{LK}{2}\eta^2 \sum_{j=0}^{n^*-1} (1 + L\eta)^j = \frac{LK}{2}\eta^2 \frac{(1 + L\eta)^{n^*} - 1}{L\eta}$$

$$\leq \frac{K\eta}{2}(1 + L\eta)^{n^*} \overset{(*)}{\leq} \frac{K\eta}{2}e^{L\eta n^*} \overset{(**)}{\leq} \frac{K\eta}{2}e^{L(R+1)} \ .$$

In the above, $(*)$ is due to the fact that for all $s \in \mathbb{R}$ it holds that $1 + s \leq e^s$, and $(**)$ is due to $\eta \cdot n^* \leq R + 1$. Now, we notice that:

$$\frac{K\eta}{2}e^{L(R+1)} \xrightarrow{\eta \to 0} 0 \ ,$$

which concludes our proof. $\qquad\qquad\square$

To apply this result in the context of optimization curves, we have the following immediate corollary:

**Corollary 1.** *Let $L > 0$, $f : \mathbb{R}^n \to \mathbb{R}$ be a convex $L$-smooth function and $x_0 \in \mathbb{R}^n$. For every $\eta > 0$ let $x^{(\eta)}(\cdot)$ be the corresponding Euler's approximation, and let $x(\cdot)$ be the unique solution for the corresponding gradient flow differential equation. Then for all $t \geq 0$:*

$$f\left(x^{(\eta)}(t)\right) \xrightarrow{\eta \to 0} f\left(x(t)\right)$$

*Proof.* Theorem 7 obviously implies that for all $t \geq 0$ we have:

$$x^{(\eta)}(t) \xrightarrow{\eta \to 0} x(t) \ .$$

The result now follows from the continuity of $f$. $\qquad\qquad\square$

Now, we would like to use the convergence established by this corollary, along with the results of Theorem 1, to prove that the optimization curve of gradient flow is convex as well. However, one should note that Theorem 1 establishes the convexity of the optimization curve of *gradient descent* (which is defined to be

the linear interpolation between the points $\{(n, f(x_n))\}_{n=0}^{\infty})$, rather than the convexity of the mapping $t \mapsto f(x^{(\eta)}(t))$ discussed in the corollary. Therefore, we shall need a variant of Theorem 1, which concerns the convexity of the mapping $t \mapsto f(x^{(\eta)}(t))$. For technical reasons, we will utilize a rather different proof technique to prove the result. The aforementioned variant is formalized as follows:

**Theorem 8.** *Let $f : \mathbb{R}^n \to \mathbb{R}$ be a convex $L$-smooth function, and let $x_0 \in \mathbb{R}^n$. For every $\eta > 0$ let $x^{(\eta)}(\cdot)$ be Euler's approximation of the corresponding gradient flow differential equation with step size $\eta$. Then for all $\eta \in (0, \frac{1}{L}]$ the map $t \mapsto f\left(x^{(\eta)}(t)\right)$ is convex (over $[0, \infty)$).*

*Proof.* By the definition of $L$-smoothness, if follows that $f$ is also $L'$-smooth for all $L' \geq L$. Therefore, for all $\eta \in (0, \frac{1}{L}]$ it holds that $f$ is $\frac{1}{\eta}$-smooth. Therefore, without loss of generality, it is sufficient to prove the result for $\eta = \frac{1}{L}$.

Looking at $x^{(\frac{1}{L})}$, it is easy to notice by its definition that for all $n \in \mathbb{Z}_{\geq 0}$, $x^{(\frac{1}{L})} \in C^1([\frac{n}{L}, \frac{n+1}{L}], \mathbb{R}^n)$. Thus, it holds that $x^{(\frac{1}{L})} \in C^1_{PW}([0, \infty), \mathbb{R}^n)$ - $x^{(\frac{1}{L})}$ is a piece-wise $C^1$ curve on $[0, \infty)$. Thus, by the chain rule and the fact that $f \in C^1(\mathbb{R}^n, \mathbb{R})$, it holds that the mapping $t \mapsto f\left(x^{(\frac{1}{L})}(t)\right)$ is in $C^1([\frac{n}{L}, \frac{n+1}{L}], \mathbb{R})$ for every $n \in \mathbb{Z}_{\geq 0}$, and thus, it is a $C^1$ piece-wise function on $[0, \infty)$.

Now, define $g : [0, \infty) \to \mathbb{R}$ as

$$g(t) = \left\langle \nabla f\left(x^{(\frac{1}{L})}(n_t/L) - (t - n_t/L)\nabla f\left(x^{(\frac{1}{L})}(n_t/L)\right)\right), -\nabla f\left(x^{(\frac{1}{L})}(n_t/L)\right) \right\rangle,$$

where we denote $n_t = \lfloor tL \rfloor$. It is easily verified that wherever the map $t \mapsto f\left(x^{(\frac{1}{L})}(t)\right)$ is differentiable, its derivative is exactly $g(t)$. Considering the fact that this mapping is differentiable at $[0, \infty) \setminus \{\frac{n}{L}\}_{n=0}^{\infty}$ and that $g$ is easily seen to be a piece-wise continuous function (and thus it is Riemann integrable on every closed sub-interval of $[0, \infty)$), we have by the Newton-Leibniz theorem (see the exact form in Bartle & Sherbert (2011, Theorem 7.3.1)) that

$$f\left(x^{(\frac{1}{L})}(t)\right) - f(x_0) = \int_0^t g(s)ds$$

for all $t \geq 0$. Now, by lemma 1, proving that $g$ is non-decreasing shall suffice, as it shall imply that the map $t \mapsto f\left(x^{(\frac{1}{L})}(t)\right)$ is convex over $[0, \infty)$.

To that end, we shall prove the following:

  (i) For all $n \in \mathbb{Z}_{\geq 0}$, $g$ is non decreasing on $[\frac{n}{L}, \frac{n+1}{L})$.

  (ii) For all $n \in \mathbb{N}$, $\lim_{t \to \frac{n}{L}^-} g(t) \leq g(\frac{n}{L})$.

Along with the fact that $g$ is piece-wise continuous (with discontinuities at $\{\frac{n}{L}\}_{n=0}^{\infty})$, these two properties are easily seen to imply that $g$ is non-decreasing. Thus, proving them will conclude the proof.

Thus, we now turn to prove (i). Let there be $n \in \mathbb{Z}_{\geq 0}$ and let there be $t_1, t_2 \in [\frac{n}{L}, \frac{n+1}{L})$ such that $t_2 > t_1$. Denote $y(t) = x^{(\frac{1}{L})}\left(\frac{n}{L}\right) - \left(t - \frac{n}{L}\right)\nabla f\left(x^{(\frac{1}{L})}\left(\frac{n}{L}\right)\right)$ for $t \in [\frac{n}{L}, \frac{n+1}{L})$. Then

$$(t_2 - t_1)(g(t_2) - g(t_1)) = \langle \nabla f(y(t_2)) - \nabla f(y(t_1)), y(t_2) - y(t_1) \rangle \geq 0,$$

where the inequality follows from convexity of $f$ (and the fact that it is in $C^1(\mathbb{R}^n, \mathbb{R})$). It obviously follows that $g(t_2) \geq g(t_1)$, as desired.

Turning to prove (ii), let there be $n \in \mathbb{N}$ and denote $z_0 = x^{(\frac{1}{L})}\left(\frac{n-1}{L}\right)$, $z_1 = x^{(\frac{1}{L})}\left(\frac{n}{L}\right)$. Obviously we have:

$$\lim_{t \to \frac{n}{L}^-} g(t) = \left\langle \nabla f\left(z_0 - \frac{1}{L}\nabla f(z_0)\right), -\nabla f(z_0) \right\rangle.$$

We now notice the following:

$$
\begin{aligned}
g\left(\frac{n}{L}\right) - \left(\lim_{t\to\frac{n}{L}^-} g(t)\right) &= \langle \nabla f(z_1), -\nabla f(z_1)\rangle - \langle \nabla f(z_1), -\nabla f(z_0)\rangle = \langle \nabla f(z_1), \nabla f(z_0) - \nabla f(z_1)\rangle = \\
&= \langle \nabla f(z_1), \nabla f(z_0) - \nabla f(z_1)\rangle - \langle \nabla f(z_0), \nabla f(z_0) - \nabla f(z_1)\rangle + \langle -\nabla f(z_0), \nabla f(z_1) - \nabla f(z_0)\rangle = \\
&= \langle \nabla f(z_1) - \nabla f(z_0), -\nabla f(z_0)\rangle - \|\nabla f(z_1) - \nabla f(z_0)\|^2 = \\
&= L\left(\left\langle \nabla f(z_1) - \nabla f(z_0), -\frac{1}{L}\nabla f(z_0)\right\rangle - \frac{1}{L}\|\nabla f(z_1) - \nabla f(z_0)\|^2\right) = \\
&= L\left(\langle \nabla f(z_1) - \nabla f(z_0), z_1 - z_0\rangle - \frac{1}{L}\|\nabla f(z_1) - \nabla f(z_0)\|^2\right) \geq 0 \; ,
\end{aligned}
$$

where the inequality holds since $f$ is convex and $L$-smooth (see (Nesterov, 2018, Theorem 2.1.5)). Thus, $\lim_{t\to\frac{n}{L}^-} g(t) \leq g(\frac{n}{L})$ as desired. As mentioned above, this concludes the proof.

$\square$

Now, having these results, we just need one more technical lemma in order to prove theorem 5.

**Lemma 2.** *Let $A \subset \mathbb{R}^n$ be a convex set, $f : A \to \mathbb{R}$ and $g : (0,\infty) \times A \to \mathbb{R}$ such that:*

1. *For every $x \in A$: $g(s,x) \xrightarrow{s\to 0} f(x)$.*

2. *There exists $\delta > 0$ such that for all $s \in (0,\delta)$ the map $x \mapsto g(s,x)$ is convex over $A$.*

*Then $f$ is a convex function.*

*Proof.* Fix some $x, y \in A$ and $\lambda \in (0,1)$. Then for every $s \in (0,\delta)$, by convexity of $x \mapsto g(s,x)$, it holds that

$$g(s, \lambda x + (1-\lambda)y) \leq \lambda g(s,x) + (1-\lambda)g(s,y) \; .$$

Now, taking the limit $s \to 0$ in both sides of the above inequality, and using the first assumption in the lemma statement, we get

$$f(\lambda x + (1-\lambda)y) \leq \lambda f(x) + (1-\lambda)f(y).$$

Since $x, y, \lambda$ are arbitrary, the result follows by definition of convexity. $\square$

Now we are ready to prove theorem 5.

*Proof of Theorem 5.* As usual, for every $\eta > 0$ we denote the corresponding Euler's approximation as $x^{(\eta)}(\cdot)$. By theorem 8 we get that for every $\eta \in (0, \frac{1}{L})$, the map $t \mapsto f\left(x^{(\eta)}(t)\right)$ is convex over $[0,\infty)$. Corollary 1 implies that for every $t > 0$,

$$f\left(x^{(\eta)}(t)\right) \xrightarrow{\eta\to 0} f\left(x(t)\right) \; .$$

Combined with Lemma 2, we get that the map $t \mapsto f\left(x(t)\right)$ is convex over $[0,\infty)$, as required. $\square$

