# OpenReview forum: "Are Convex Optimization Curves Convex?"
_TMLR — Accepted by TMLR_

### Review · Reviewer_MKfE · 2025-04-10

**Summary Of Contributions:**

The paper studies whether the iteration/function value curves of gradient descent (GD) and gradient flow (GF) are convex. In particular, it examines whether the sets $\{n, f(x_n)\}$ (piecewise linearly connected) and $\{t, f(x(t))\}$ have convex epigraphs. The authors provide different results depending for GF and GD for convex and $L$-smooth objectives:

- **GD with step size $\eta$**: When $\eta \in [0, 1.75/L]$, the iteration curve is convex.
- **GD with step size $\eta$**: Although GD is convergent for $\eta < 2/L$, in the range $\eta \in [1.75/L, 2/L]$, the authors provide a counterexample showing that the iteration curve is neither convex nor concave.
- **GF**: For convex and smooth functions, the curve is convex.

**Audience:**

Yes

**Broader Impact Concerns:**

The paper is of theoretical nature. There are no ethical concerns.

**Claims And Evidence:**

Yes

**Requested Changes:**

### Mathematical

- It would be helpful to provide a geometric definition of what is meant by convexity of the curve, possibly using epigraphs to formalize it.
- **Proof of Theorem 6**: The derivative of $g$ should be $-|| \nabla f ||^2$. This does not affect the result, but should be corrected for accuracy.
- Theorem 6 and Theorem 3 show that the gradient norms of GF and GD are non-increasing. However, I assume that these results can be found in prior work, although some off-the-shelf references did not have it.

### Non-Mathematical

- **Page 3** – `lemma 1` → `Lemma 1`
- **Page 4** – "`Rearranging terms, left-hand side equals`" → "`Rearranging terms, right-hand side equals`"
- **Page 10** – Missing period in the last equation of Corollary 1.
- **Page 11** – "`and thus, a $C^1$ function...`" → "`and thus, $f$ is a $C^1$ function...`"
- **Page 11** – The definition of the function $g$ is slightly bulky. It would be clearer to use different sizes of parentheses to improve readability.

**Strengths And Weaknesses:**

The paper addresses an interesting problem in geometry and calculus. Reading the paper is a pure mathematical joy. The techniques are simple yet clever, and the proofs are correct.
The paper definitely deserves to be accepted to TMLR.

---

> ### Author Response · Authors · 2025-05-19
> **Responses to Reviewer MKfE**
>
> Thanks for your review! See response below.
> ### Mathematical
> 1. **Convexity of the curve:** We added some clarification and a mention of epigraphs in the preliminaries section. We also provide a geometric intuition in terms of the slope of the curve after the formal definition.
> 2. **Proof of thm. 6:** Fixed.
> 3. **Prior work on gradient norms being non-increasing:** We were unable to locate these particular results in prior work. Of course, if there is a relevant reference, we would be glad to add it.
>
> ### Non-Mathematical
> 1. Fixed
> 2. Fixed
> 3. Fixed
> 4. Fixed
> 5. We tried to make the expression easier to parse

---

### Review · Reviewer_SbwR · 2025-04-21

**Summary Of Contributions:**

The paper studies when the trajectory of gradient descent results in a convex function on a smooth and convex function. For gradient descent (GD) $x_{n+1}=x_n - \eta \nabla f(x_n)$ where $\eta$ is the stepsize and $f$ is $L$-smooth and convex, standard analysis tells that it converges monotonically when $\eta\leq2/L$. However, little is known whether the trajectory $(n, f(x_n))$ is convex. The contribution can be summarized below.

1. When $\eta\leq1.75/L$, the trajectory of GD is always convex. When $\eta\in(1.75,2)/L$, although GD always converges with $f(x_n)$ monotonically decreasing, there exist examples where the trajectory of GD is not convex.

2. When considering the $\Vert\nabla f\Vert$, the trajectory is always non-increasing.

3. These two results are extended to gradient flow when $\eta\to0$. The trajectory $(t, f(x_t))$ is always convex, and the trajectory $(t, \Vert\nabla f(x_t)\Vert)$ is non-increasing.

**Audience:**

Yes

**Broader Impact Concerns:**

Not applied. No ethical concerns.

**Claims And Evidence:**

Yes

**Requested Changes:**

My biggest concern is on the motivation of studying the considered problem.

1. Optimal algorithms, AGD, for smooth convex functions do not produce convex curves, but they achieve the best convergence rate.

2. The only example of convex curves is GD applied to smooth convex functions.

It seems that whether curves are convex is not super relevant to whether the algorithm is good or not. There are not too many examples where we can guarantee convex curves as well. The question why we should look at convexity of optimization curves is not clear. It is also not motivated enough why we should focus on non-decreasing gradient norm as well. I mentioned one alternative to study what stepsize overshoots. This appears to be a more relevant and interesting problem for me.

I hope the points I mentioned in the weakness section and this biggest concern I summarized above could be resolved in the next version.

**Strengths And Weaknesses:**

**Weaknesses**:

1. I think the settings where the trajectory is convex are very limited. I guess it is only possible to guarantee GD on deterministic smooth and convex functions. This property easily breaks down if considering stochastic setting with gradient noise, nonconvex functions, nonsmooth functions, accelerated GD, etc.

2. The case where the trajectory is not convex is not necessarily bad. For example, GD/SGD works for minimizing deep neural networks, where the trajectory is likely not convex. For the smooth and convex function this paper considers, although GD achieves convex curves, GD is not the optimal algorithm for solving it. The optimal algorithm, accelerated GD, adds some extrapolation/momentum step and likely breaks down the convex curves, but it brings faster convergence. Based on 1 and 2, the problem to study whether curves are convex seems to be meaningful and interesting only in math, it does not necessarily result in anything. Nonconvex curves do not necessarily hurt anything either.

3. All examples where convex curves fail in this paper are caused by overshooting. For this, I mean the trajectory passes the optimal point. For Example 1, $x_0=-\eta/4$ and $x_1=3\eta/4$, and the curve from $x_0$ to $x_1$ passes the critical point $0$. The example in Theorem 2 has $x_0=-1.8$ and $x_1=1.8(\eta-1)$, which passes by the critical point $0$. Therefore, it seems to me a more interesting problem is to see which stepsize does not overshoot. The same motivation and the same set of results still work, but it allows more instances. Overshooting is also hurting more as it causes oscillations and unnecessary additional optimization steps.

4. It is not fully clear to me why studying curves for gradient norm is related. At first glance, a more meaningful result would be also studying whether the curve is convex or not instead of just non-increasing according to the title. Although there are some links between convex curves of loss and non-increasing curves of gradients similarly to Lemma 1, it is not fully clear in the high-dimensional case. Also, GD is not optimal in making gradients small. The (near) optimal algorithms "GD after AGD" and "AGD with regularization"discussed in Nesterov's book not necessarily results in non-increasing curves as well.

**Other questions** (These are questions (potential future work) I think about when reading the paper. These are not necessarily something claimed or studied by the paper):

1. The current paper looks at fixed constant stepsize. What happens if the stepsize varies? I think it is easy to design a diminishing stepsize where it converges with a convex curve. Also, there is a recent line of work studying acceleration of GD using large stepsize [1,2], where the stepsize varies.

2. Quotation marks in latex is produced using `` and ''.

3. Why does the following equation on page 6 (after equation (7)) hold? We can choose $\eta=0.5/L$ and the first term is negative, so it is not obvious to me why this is always true when $\Vert\nabla f(x_1)\Vert>0$.

$$(\eta - 1/L) \Vert \nabla f(x_0)\Vert + 1/L\Vert \nabla f(x_1)\Vert >0$$

4. The paper said on page 6 that the gradient flow has a unique global solution (before Definition 2). What does global solution mean?

5. As the paper also mention in Section 5, I think it is more interesting to study examples beyond convex smooth functions. To me, convexity of curves is not super interesting. It is good to also think about other behaviours. For example, in low rank matrix factorization $\Vert XX^\top - A\Vert_F^2$ problems, as it contains saddle-point and many local minima, the curves often behave like step-wise functions. This might be a suitable prototype to study optimization trajectories.

**References:**

[1] Provably faster gradient descent via long steps. SIAM OPT. 2024.

[2] Acceleration by stepsize hedging: Silver Stepsize Schedule for smooth convex optimization. Math.Program. 2024.

---

> ### Author Response · Authors · 2025-05-19
> **Responses to Reviewer SbwR**
>
> Thanks for your review! See response below.
> ### Weaknesses
> 1. **Limitations of the setting:** As discussed in the paper, we believe that the question of when might we expect  convexity of the optimization curve (as well as monotonic decrease of the gradient norm) are of interest, as these are fundamental theoretical properties of the optimization process, which have not been studied at all so far. In addition, it may have interesting applications, e.g. in the context of understanding when optimization plateaus may be avoided. We do agree that one may form different opinions about the immediate applicability of these theoretical questions. However, since TMLR's scope is specifically to *“emphasize technical correctness over subjective significance”* (see [https://jmlr.org/tmlr/](https://jmlr.org/tmlr/)), we believe this should not be a major concern here.
> 2. **Non-convex curves not always harmful:** See response to item 1 above.
> 3. **Bad behavior caused by overshooting:** We agree, but in our view it is interesting to show that convexity does not occur even in the regime long-known to result in convergence. Understanding what can happen without overshooting is an excellent question for future work.
> 4. **Relation between optimization curve convexity and gradient norm decay:** We agree it is not quite the same question (indeed, our results in terms of the step sizes regimes are different for each one). However, there is a connection between the two, and we added a discussion about this point in the last paragraph of Section 1.
>
> ### Other Questions
> 1. **Varying step-sizes:** Indeed, that's an excellent question for future work.
> 2. **Quotation marks:** Fixed.
> 3. **Equation on pg. 6:** We added an explanation.
> 4. **What does global solution mean:** Sorry for the confusing terminology; by “global solution” we meant a unique, well-defined solution globally on all of $t \geq 0 $. We changed the phrasing to make this clearer.
> 5. **Other problems:** We agree it will be interesting to study such problems in future work.

---

### Review · Reviewer_UXUq · 2025-05-02

**Summary Of Contributions:**

This paper proposes an interesting notion of "convexity of an optimization curve"—a sequence of numbers $\set{ v_i }$ is called convex if the linear interpolation of the sequence $\set{ ( i, v_i ) }$ is convex. Consider minimizing a function using gradient descent with a constant step size. The paper proves the following results:
1. If the function is convex and $L$-smooth, then the corresponding optimization curve is convex for step sizes between $0$ and $1.75 L$, and may become non-convex if the step size exceeds $1.75 L$.
2. If the function is convex and $L$-smooth, then the gradient norm of the objective function is non-increasing along the optimization curve.
3. If the function is non-convex and Lipschitz continuous, then the corresponding optimization curve can be non-convex for any constant step size.

The paper also proves that for gradient flow applied to a convex function, the optimization curve remains convex and the gradient norm is also non-increasing.

**Audience:**

Yes

**Broader Impact Concerns:**

I believe there is no such concern, as this is a theoretical work.

**Claims And Evidence:**

Yes

**Requested Changes:**

1. Please try to find a counter example for the non-convex case that is smooth.
2. Please try to argue for the significance of Theorem 2. It might be helpful to provide a counterexample involving a commonly used loss function in machine learning.
3. Please clarify the connection between Theorem 3 and the title.
1. In the proof of Theorem 6, please either remove the uninformative phrase “by elementary results” or replace it with something more specific, such as “by the monotonicity of gradients of convex functions.”

**Strengths And Weaknesses:**

**Strengths.**
1. The problem formulation is interesting. The notion of "convexity of an optimization curve" is novel, to the best of my knowledge.
2. I appreciate the authors' effort to provide a complete solution to when the optimization curve of gradient descent is convex for the standard convex and smooth case.
2. The proofs are simple and easy to check.

**Weaknesses.**
1. The paper motivates this study by stating that the convexity of an optimization curve is sufficient to prevent the “plateau phenomenon.” This motivation is somewhat weak, given the obvious fact that convexity is not a necessary condition. It is unclear whether convexity is the appropriate notion to study.
2. The “plateau phenomenon” is perhaps more relevant in the context of non-convex optimization. Nevertheless, the authors exclude the non-convex case using a counterexample (Example 1), which corresponds to the third result I mentioned above. The counterexample is somewhat weak, as the paper primarily considers smooth functions, whereas the counterexample is Lipschitz continuous and non-smooth. Strictly speaking, the counterexample does not rule out the possibility that the optimization curve can be convex in non-convex smooth optimization.
3. The authors use another counterexample to show that the convexity of the optimization curve does not generally hold for step sizes exceeding $1.75 L$ (Theorem 2). However, since only one counterexample is provided, the significance of this negative result remains unclear.
3. The second result I mentioned above, which establishes the monotonicity of the gradient norm along the optimization curve (Theorem 3), is not closely related to the title.
4. The results for gradient flow are not very interesting. Gradient flows are not implementable, and the step size selection issue does not arise in this case.

---

> ### Author Response · Authors · 2025-05-19
> **Responses to Reviewer UXUq**
>
> Thanks for your review! See response below.
> ### Weaknesses
> 1. **Motivation for the setting:** As discussed in the paper, we believe that the question of the convexity of the optimization curve (as well as monotonic decrease of the gradient norm) are of interest, as these are fundamental theoretical properties of the optimization process, which have not been studied at all so far. In addition, it may have interesting applications, e.g. in the context of understanding when optimization plateaus may be avoided. We do agree that one may form different opinions about the immediate applicability of these theoretical questions. However, since TMLR's scope is specifically to *“emphasize technical correctness over subjective significance”* (see [https://jmlr.org/tmlr/](https://jmlr.org/tmlr/)), we believe this should not be a major concern here.
> 2. **Nature of the counterexample:** There may have been some misunderstanding here. Example 1 refers to a convex, non-smooth function (not a non-convex function as mentioned in the review). The case of non-convex, smooth optimization is discussed in the paragraph preceding Example 1 (in that case, it is easily seen that the optimization curve will not be convex in general).
> 3. **Convexity does not always hold:** The purpose of the counterexample is to show that the optimization curve will not always be convex beyond a certain step size. To establish such a result, it is enough to produce a single counterexample. It would indeed be interesting in the future to characterize more generally when convexity may not hold, but it is not the focus of our current paper.
> 4. **Connection between convexity of optimization curve and gradient norm decay:** We added a discussion of this point in the last paragraph of Section 1.
> 5.  **Results for gradient flow:** Gradient flow is a very well-studied abstraction of gradient descent. As to subjective significance, see response to item 1 above.
>
> ### Changes
> 1. See response to item 2 above.
> 2. See response to item 2 above.
> 3. See response to item 4 above.
> 4. Thanks for the good suggestion, fixed.

---

> ### Comment · Reviewer_UXUq · 2025-06-09
>
> **Nature of the counterexample**: Yes, I made a mistake regarding Example 1, which was introduced to exclude convex but non-smooth functions. Thank you for pointing it out.
>
> **Convexity does not hold**: It is certainly correct that a single counterexample is sufficient to refute a claim. Nevertheless, one counterexample may not be sufficient to establish the significance of a claim. I would suggest that the authors reconsider my second suggestion.

---

### Comment · Action_Editor_c6B1 · 2025-05-16
**Any response to the reviews?**

Dear authors,

It's been a few weeks since you received the reviews, and some of them requested changes and pointed out weaknesses. If you're planning to address them in your response, please do it soon.

Your action editor

---

### Decision · Action_Editor_c6B1 · 2025-06-18

**Recommendation:** Accept as is

**Audience:**

Yes

**Audience Explanation:**

The paper received a very positive feedback from the reviewers, the problem is clear and the paper is enjoyable to read.

**Claims And Evidence:**

Yes

**Claims Explanation:**

The paper is very rigorous, there were small issues pointed out by the reviewers, but the quality of this work is high overall.